

# Complete mitochondrial genome sequence of the "copper moss" *Mielichhoferia elongata* reveals independent *nad7* gene functionality loss

Denis V. Goryunov[1], Svetlana V. Goryunova[2], Oxana I. Kuznetsova[3], Maria D. Logacheva[1], Irina A. Milyutina[1], Alina V. Fedorova[1], Michael S. Ignatov[3] and Aleksey V. Troitsky[1]

[1] Belozersky Institute of Physico-Chemical Biology, Lomonosov Moscow State University, Moscow, Russia
[2] Institute of General Genetics Russian Academy of Science, Moscow, Russia
[3] Tsitsin Main Botanical Garden Russian Academy of Science, Moscow, Russia

## ABSTRACT

The mitochondrial genome of moss *Mielichhoferia elongata* has been sequenced and assembled with Spades genome assembler. It consists of 100,342 base pairs and has practically the same gene set and order as in other known bryophyte chondriomes. The genome contains 66 genes including three rRNAs, 24 tRNAs, and 40 conserved mitochondrial proteins genes. Unlike the majority of previously sequenced bryophyte mitogenomes, it lacks the functional *nad7* gene. The phylogenetic reconstruction and scrutiny analysis of the primary structure of *nad7* gene carried out in this study suggest its independent pseudogenization in different bryophyte lineages. Evaluation of the microsatellite (simple sequence repeat) content of the *M. elongata* mitochondrial genome indicates that it could be used as a tool in further studies as a phylogenetic marker. The strongly supported phylogenetic tree presented here, derived from 33 protein coding sequences of 40 bryophyte species, is consistent with other reconstructions based on a number of different data sets.

## INTRODUCTION

Bryophytes (mosses, liverworts, and hornworts) represent an ancient group of higher plant evolution that shows a dominance of the gametophyte stage in the life cycles. These nonvascular pioneers of land plants first acquired adaptations that enabled the transition from aquatic to terrestrial habitats. Mosses (Bryophyta), branched off from the stem of the Embryophyta phylogenetic tree after the Marchantiophyta and before the separation of the Anthocerotophyta (*Liu et al., 2014*; *Qiu et al., 2006*).

The mitogenomes (MGs) of mosses have recently become a target of sequencing efforts for phylogenetic reconstructions due to their compact size and a higher degree of synteny than is observed in vascular plants (*Liu, Medina & Goffinet, 2014*). The NCBI RefSeq

Corresponding authors
Denis V. Goryunov,
denis.goryunov@mail.ru
Aleksey V. Troitsky,
bobr@belozersky.msu.ru

database (http://www.ncbi.nlm.nih.gov/refseq) currently contains 39 mitochondrial genomes for members of this group of plants. This quite limited data set includes only representatives of nine orders from the Bryopsida and three orders from other classes of mosses, so it clearly does not perfectly reflect bryophyte diversity. The aim of the present study is to extend bryophyte taxonomical coverage and expand the phylogenetic analysis to include MG data from the still unexplored order Bryales. For this purpose the complete MG of *Mielichhoferia elongata* (Hoppe & Hornsch.) Nees & Hornsch. was sequenced.

The plant list (http://www.theplantlist.org, Version 1.1 September 2013) contains 142 accepted species names of *Mielichhoferia* Nees & Hornsch. The taxonomical status of *Mielichhoferia* remains under debate. The genus has usually been treated as the subfamily Mielichhoferioideae within the Bryaceae, although some authors have attributed it to a separate family, the Mielichhoferiaceae (*Hill et al., 2006*; *Shaw, 2014*). The only molecular study of phylogenetic relationships of *Mielichhoferia* placed the Mielichhoferiaceae within the Mniaceae according to the *trnL-F* and *rps4* sequence data (*Guerra et al., 2011*).

Several moss and hepatic species are restricted to substrates enriched in heavy metals. These bryophytes that show an affinity for metalliferous substrates have been referred to as "copper mosses" (*Antonovics, Bradshaw & Turner, 1971*; *Chopra & Kumra, 1988*; *Persson, 1948*, *1956*; *Shaw, 1987*, *1989*). *M. elongata* Homsch. (*Shaw, 2000*) and the closely related *Mielichhoferia mielichhoferiana* (Funck.) Loeske are among the species that are highly tolerant and largely restricted to substrates enriched in copper. These species are widely distributed around the globe, but are always rare. They grow in habitats rich in copper (often associated with other metals) and inorganic sulfides, which results in a very low pH. These habitats represent areas damaged by mining (mine waste tailings) or metal-rich rocks. The heavy metal tolerance mechanisms are not well understood and apparently vary across species. Metals are adsorbed by the cell walls and are accumulated in cells (*Antonovics, Bradshaw & Turner, 1971*; *Antreich, Sassmann & Lang, 2016*; *Brown, 1982*; *Meharg, 2005*; *Tyler, 1990*).

Knowledge of the sequence of the mitochondrial genome of *M. elongata* will be useful both for finding an appropriate taxonomic treatment for the taxa and for population studies within the Mielichhoferia. The latter studies are particularly important in light of the disruptive character of the habitat area, the rarity of these species, and ongoing habitat damage.

## MATERIALS AND METHODS

### Sample collection and DNA isolation

The *M. elongata* samples were collected from July 12 to 17, 2011 in the area near Mus-Khaya Peak (62°31′–36′N, 140°56′–141°07′E), Republic of Sakha (Yakutia) and deposited in MHA, the Herbarium of the Main Botanical Garden Russian Academy of Science, Moscow (*Ignatova et al., 2011*). This moss was originally identified in a cited paper as *M. mielichhoferiana*. However, a subsequent analysis of the nuclear rDNA 5.8S-ITS 2 region attributed this plant to that of morphologically hardly distinguishable

*M. elongata* (Fig. S1). Rocks in the area are especially rich in MnS, with other heavy metals (Pb, Sn, As, Zn, Ag, etc., usually as sulfides) present in high concentrations. Consequently, many brooks have very acidic water and sulfur deposits along them. Siderite (iron carbonate) forms red outcrops ("Red rocks") rich in iron and is always enriched with other heavy metals. When the outcrops are dry, *M. elongata* is the only moss that grows on this substrate or at least it is the only particularly abundant one.

DNA was extracted from specimens in the herbarium collection that had been gathered with a minimal soil amount and dried using ordinary herbarium techniques (in a paper envelope, under a tent, in the shade for several days until dry), and then stored in the herbarium at room temperature. A Nucleospin Plant DNA Kit (Macherey Nagel, Düren, Germany) was used for total DNA extraction from whole shoots of plants according to the manufacturers' protocol. A yield of about 2 μg DNA was obtained according to measurements determined with a Qubit fluorometer (Invitrogen, Carlsbad, CA, USA).

## Library preparation and sequencing

A 500 ng sample of genomic DNA was fragmented using a Covaris S220 sonicator (Covaris, Woburn, MA, USA) and a library was prepared using TruSeq DNA sample preparation kit (Illumina, Mountain View, CA, USA). The concentration of the prepared library was measured with the Qubit fluorometer (Invitrogen, Carlsbad, CA, USA) and qPCR and fragment length distribution was determined with Bioanalyzer 2100 (Agilent, Santa Clara, CA, USA). The library was diluted to 10 pM and used for cluster generation on a cBot instrument with TruSeq PE Cluster Kit v3 reagents (Illumina, Mountain View, CA, USA). Sequencing was performed on a HiSeq2000 sequencer with read length of 101 from both ends of the fragments. About six million read pairs were obtained.

## Mitogenome assembly and annotation

Raw sequencing reads were preprocessed with Trimmomatic software (*Bolger, Lohse & Usadel, 2014*) to remove adapters and low-quality data from further analysis. The whole genome assembly was then accomplished using the Spades assembler (*Bankevich et al., 2012*). A Blast database was generated from the assembled contigs, and a Blast search was performed against the *Physcomitrella patens* MG sequence (*Terasawa et al., 2007*) using the standalone NCBI BLAST-2.2.29+ (*Altschul et al., 1990*). The longest hit was the *M. elongata* complete MG. Iterative mapping was carried out using Geneious R10 software (https://www.geneious.com; *Kearse et al., 2012*) to verify the assembled genome. The resulting sequence had almost 100× coverage depth. The correctness of the genome boundaries was verified by PCR amplification followed by Sanger sequencing. Initial reads mapping to the genome sequence with Bowtie 2 (*Lingmead et al., 2009*) was applied as an additional genome structure verification step.

Genome annotation based on sequence similarity was performed using Geneious software. The MG sequence of *Bartramia pomiformis* which gave a maximum score in a BLAST search against a *M. elongata* MG query was applied as a reference. The annotated genome sequence was submitted to GenBank (accession number: MF417767). A circular

genome map was drawn using the CGView Server (*Grant & Stothard, 2008*; http://stothard.afns.ualberta.ca/cgview_server).

## SSR analysis

Simple sequence repeats (SSRs) were detected and located in the MG of *M. elongata* using GMATo v1.2 software (*Wang, Lu & Luo, 2013*).

## Phylogenomic analysis

Phylogenetic reconstruction was conducted by selecting only functional protein-coding sequences (CDS) present in MGs of all bryophytes under investigation. A total of 33 of these CDS are known, including *atp1, atp4, atp6, atp8, atp9, cob, cox1, cox2, cox3, nad1, nad2, nad3, nad4, nad4L, nad5, nad6, nad9, rpL2, rpL5, rpL6, rpL16, rps1, rps2, rps4, rps7, rps11, rps12, rps13, rps14, rps19, sdh3, sdh4, and tatC*. These were extracted from the MG sequences of 39 mosses, liverwort *Treubia lacunose* available in GenBank (http://www.ncbi.nlm.nih.gov), and the *M. elongata* sequenced in this work. The GenBank files were imported into Geneious R10 and merged to export a fasta dataset file. All sequences from this dataset were aligned using the default option implemented in MAFFT (*Katoh & Standley, 2013*). The final alignment was adjusted manually in BioEdit 7.2.5 (*Hall, 1999*).

Phylogenetic reconstruction was performed using the Bayesian method with the program MrBayes v3.2.6 (*Ronquist et al., 2012*). For Bayesian analyses, we used a parallel MPI version of MrBayes (*Altekar et al., 2004*). Two simultaneous runs of Metropolis Coupled Markov Chain Monte Carlo (MC3), both with one cold and seven heated chains were performed for 10 million generations. Two starting trees were chosen randomly. The general time reversible evolutionary model (GTR+I+G) with four rate categories was used. Posterior probabilities (PP) for trees and parameters were saved every 1,000 generations and parameters for each data partition were sampled independently from each other; the first 25% of the trees was discarded in each run. Bayesian PPs were used as branch support values.

# RESULTS

## Structure of the *M. elongata* mitogenome

The MG of *M. elongata* is 100,342 bp in length and has a typical circular structure (Fig. 1). The nucleotide composition of this genome has a GC content of 39.8%. The MG of *M. elongata* contains 66 genes including genes for three rRNAs (*rrn18, rrn26,* and *rrn5*), 24 tRNAs, and 39 conserved mitochondrial proteins (15 ribosomal proteins, four ccm proteins, eight nicotinamide adenine dinucleotide dehydrogenase subunits, five ATPase subunits, two succinate dehydrogenase subunits, one apocytochrome b, three cytochrome oxidase subunits, and one twin-arginine translocation complex subunit). Besides the functional genes, a single pseudogene, *nad7*, resides in the genome (Table 1).

## Structure of *nad7* gene in bryophytes

The lack of a functional gene copy of the *nad7* gene has been reported previously in the MG of hornworts and the majority of liverworts (*Groth-Malonek et al., 2007*;

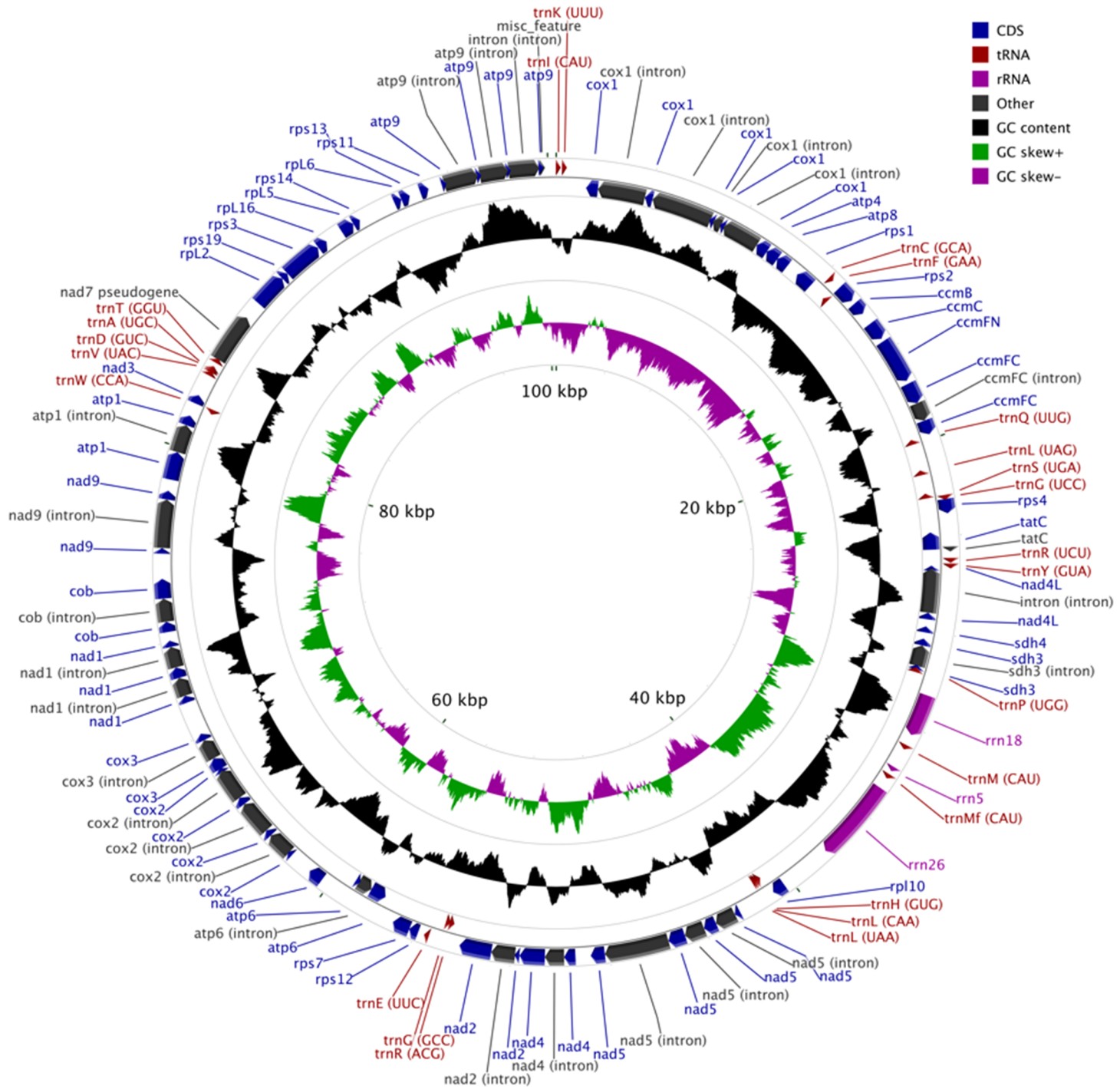

**Figure 1** **Mitogenome map of *Mielichhoferia elongata* (MF417767) consisting of 100,342 base pairs.**

*Li et al., 2009*; *Xue et al., 2010*). Evolution and losses of the functionality of the gene copies within mosses also deserve special attention and scrutiny. Pseudogenization of the *nad7* gene is currently described for *Tetraphis pellucida* and *Buxbaumia aphylla* (*Bell et al., 2014*; *Liu, Medina & Goffinet, 2014*), whereas all other sequenced bryophyte MGs have a

**Table 1 Gene contents in *Mielichhoferia elongata* mitogenome (66 genes, one pseudogene).**

| Category | Group of genes | Genes | Number of genes |
|---|---|---|---|
| RNA genes | rRNAs | *rrn18, rrn26, rrn5* | 3 |
| | tRNAs | *trnA (UGC), trnC (GCA), trnD (GUC), trnE (UUC), trnF (GAA), trnG (GCC), trnG (UCC), trnH (GUG), trnI (CAU), trnK (UUU), trnL (CAA), trnL (UAA), trnL (UAG), trnM (CAU), trnMf (CAU), trnP (UGG), trnQ (UUG), trnR (ACG), trnR (UCU), trnS (UGA), trnT (GGU), trnV (UAC), trnW (CCA), trnY (GUA)* | 24 |
| Conserved mitochondrial proteins | Large ribosomal subunits | *rpl10, rpL16, rpL2, rpL5, rpL6* | 5 |
| | Small ribosomal subunits | *rps1, rps11, rps12, rps13, rps14, rps19, rps2, rps3, rps4, rps7* | 10 |
| | Cytochrome c maturation proteins | *ccmB, ccmC, ccmFC, ccmFN* | 4 |
| | Nicotinamide adenine dinucleotide dehydrogenase subunits | *nad1, nad2, nad3, nad4, nad4L, nad5, nad6, nad9* | 8 |
| | ATPase subunits | *atp1, atp4, atp6, atp8, atp9* | 5 |
| | Succinate dehydrogenase subunits | *sdh3, sdh4* | 2 |
| | Apocytochrome b | *cob* | 1 |
| | Cytochrome oxidase subunits | *cox1, cox2, cox3* | 3 |
| | Twin arginine translocation complex subunit | *tatC* | 1 |
| Pseudogenes | | *nad7pseudo* | 1 |

functional gene, that consists of three exons separated by two introns. The only known exception is the *nad7* locus structure in MG of *Hypnum imponens* (NC 024516), its functional gene consists of only two exons and one intron sequences. The intron 2 of the gene was lost and exons 2 and 3 were merged together in one exon sequence. The low conservation of the pseudogene sequences has created difficulties in constructing a reliable nucleotide alignment and unambiguously judging whether exons 2 and 3 are completely deleted in either these chondriomes or whether some exon remnants are still preserved. We performed a Tblastn search of these exons amino acid sequence of the *nad7* gene in *T. pellucida* and *B. aphylla* and confirmed the absence of exon 2 in *B. aphylla* and exon 3 in both species. The same finding is evident from Fig. S2 with the alignment of *nad7* from *B. aphylla*, *T. pellucida*, *M. elongata*, and six other moss species. It agrees with the earlier data provided by *Bell et al. (2014)* on the structure of the *T. pellucida* MG. In addition, *B. aphylla* and *T. pellucida* pseudogenes have deletions in the sequences of the first gene exon, although at different locations. The main difference in the *nad7* pseudogene primary structure in these bryophytes is two deletions in the sequence of exon 2 in *B. aphylla*, whereas *T. pellucida* has an intact exon 2 sequence. By contrast, the *nad7* pseudogene of *M. elongata* completely lacks the second exon and has intact exon 1 sequence and exon 3 with frame shift mutation as a result of 2 bp insertion located at 190 bp from 5′ end of the exon (Fig. 2).

## SSR analysis of the *M. elongata* mitochondrial genome

Following more stringent criteria (*Zhao, Zhu & Liu, 2016*) of perfect SSR locus identification (minimal number of repeating units ≥10 for mononucleotides,

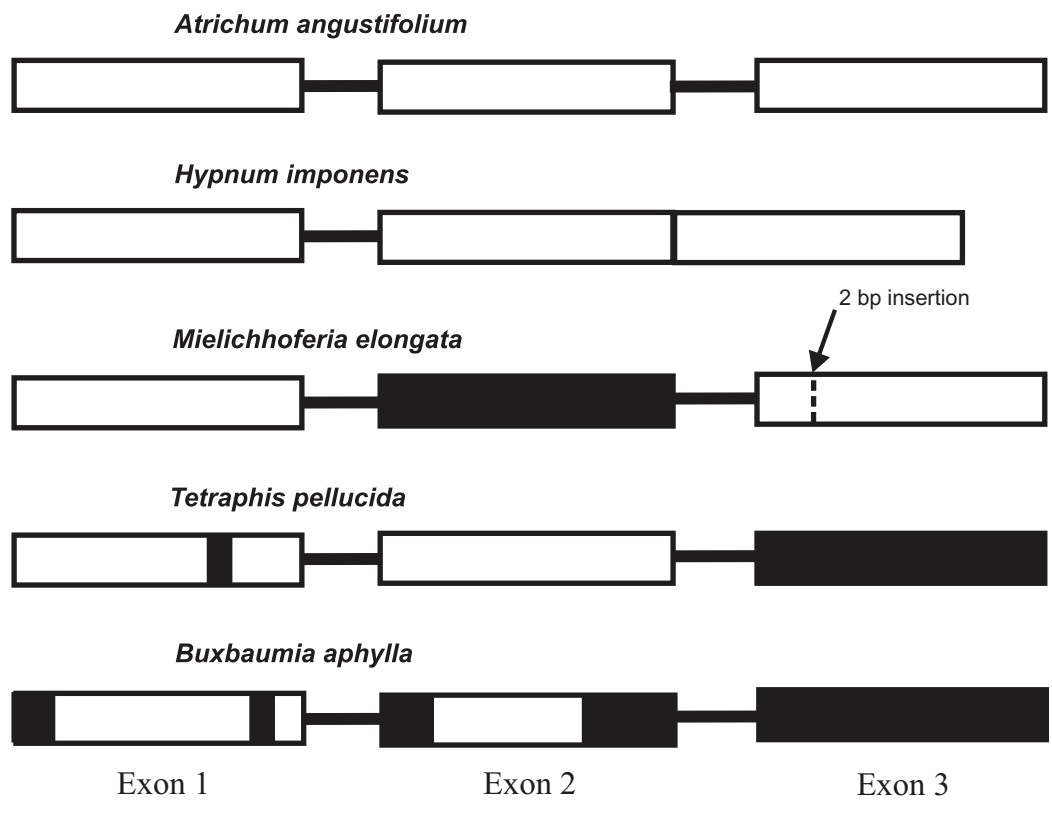

**Figure 2** **The diversity of the mitochondrial *nad7* gene exon structure in mosses.** The majority of the sequenced moss mitogenomes have the same type of locus structure found in *Atrichum angustifolium*. The black filled sections indicate absent exons (or parts of them).

≥5 for dinucleotides, ≥4 for trinucleotides, and ≥3 for tetra-, penta- and hexanucleotides) 73 SSR loci were identified in the MG of *M. elongata* (Table 2; Fig. 3). Most microsatellites refer to mono- and dinucleotides classes (35 and 28 loci, respectively). Trinucleotides are the least frequent SSRs group in the genome (one locus). No hexanucleotide microsatellite repeats occur in the genome. Among all the SSRs, 87.67% are composed only of A/T bases. The total length of the SSR loci is 852 bp, which comprises approximately 0.85% of the genome length.

## Phylogenetic analysis

The alignment of 33 mitochondrial protein CDS of 40 moss taxa and hepatic *Treubia lacunosa* (Haplomitriopsida, Treubiidae, Treubiales, Treubiaceae) consists of 24,827 positions. The Bayesian phylogenetic tree inferred from this data with the hepatic *T. lacunosa* as an outgroup is shown in Fig. 4. Most nodes of the tree have very high PP supports. Two exceptions are two nodes among the Orthotrichaceae.

## DISCUSSION

We performed sequencing and analysis of the MG of *M. elongata,* a rare "copper moss" with an ambiguous taxonomic status. The MG size significantly varies even among closely

**Table 2  SSR-loci of *Mielichhoferia elongata* mitogenome.**

| Type of repeat unit | Motif | Repetitions | StartPos | EndPos |
| --- | --- | --- | --- | --- |
| Mono- | A | 10 | 269 | 278 |
| Mono- | A | 10 | 13,526 | 13,535 |
| Mono- | A | 10 | 22,179 | 22,188 |
| Mono- | A | 10 | 25,861 | 25,870 |
| Mono- | A | 10 | 40,307 | 40,316 |
| Mono- | A | 10 | 46,592 | 46,601 |
| Mono- | A | 10 | 49,092 | 49,101 |
| Mono- | A | 10 | 52,217 | 52,226 |
| Mono- | A | 10 | 54,565 | 54,574 |
| Mono- | A | 10 | 62,618 | 62,627 |
| Mono- | A | 10 | 88,341 | 88,350 |
| Mono- | A | 10 | 91,128 | 91,137 |
| Mono- | A | 10 | 93,879 | 93,888 |
| Mono- | A | 11 | 39,390 | 39,400 |
| Mono- | A | 12 | 16,182 | 16,193 |
| Mono- | G | 10 | 98,368 | 98,377 |
| Mono- | G | 12 | 52,784 | 52,795 |
| Mono- | G | 12 | 57,418 | 57,429 |
| Mono- | T | 10 | 29,873 | 29,882 |
| Mono- | T | 10 | 46,703 | 46,712 |
| Mono- | T | 10 | 47,865 | 47,874 |
| Mono- | T | 10 | 56,400 | 56,409 |
| Mono- | T | 10 | 57,552 | 57,561 |
| Mono- | T | 10 | 86,976 | 86,985 |
| Mono- | T | 10 | 94,469 | 94,478 |
| Mono- | T | 10 | 99,146 | 99,155 |
| Mono- | T | 11 | 16,200 | 16,210 |
| Mono- | T | 11 | 25,885 | 25,895 |
| Mono- | T | 11 | 40,958 | 40,968 |
| Mono- | T | 11 | 50,459 | 50,469 |
| Mono- | T | 11 | 58,416 | 58,426 |
| Mono- | T | 11 | 95,793 | 95,803 |
| Mono- | T | 12 | 17,608 | 17,619 |
| Mono- | T | 12 | 100,200 | 100,211 |
| Mono- | T | 15 | 11,233 | 11,247 |
| Di- | AT | 5 | 32,938 | 32,947 |
| Di- | AT | 5 | 54,921 | 54,930 |
| Di- | AT | 6 | 14,278 | 14,289 |
| Di- | AT | 6 | 14,298 | 14,309 |
| Di- | AT | 6 | 59,230 | 59,241 |
| Di- | AT | 7 | 70,407 | 70,420 |

| Type of repeat unit | Motif | Repetitions | StartPos | EndPos |
|---|---|---|---|---|
| Di- | TA | 5 | 195 | 204 |
| Di- | TA | 5 | 279 | 288 |
| Di- | TA | 5 | 466 | 475 |
| Di- | TA | 5 | 27,730 | 27,739 |
| Di- | TA | 5 | 41,770 | 41,779 |
| Di- | TA | 5 | 44,628 | 44,637 |
| Di- | TA | 5 | 62,826 | 62,835 |
| Di- | TA | 5 | 68,954 | 68,963 |
| Di- | TA | 5 | 69,190 | 69,199 |
| Di- | TA | 6 | 12,557 | 12,568 |
| Di- | TA | 6 | 86,244 | 86,255 |
| Di- | TA | 6 | 94,457 | 94,468 |
| Di- | TA | 7 | 10,767 | 10,780 |
| Di- | TA | 7 | 19,813 | 19,826 |
| Di- | TA | 7 | 25,871 | 25,884 |
| Di- | TA | 7 | 28,304 | 28,317 |
| Di- | TA | 7 | 29,340 | 29,353 |
| Di- | TA | 7 | 41,786 | 41,799 |
| Di- | TA | 8 | 57,533 | 57,548 |
| Di- | TA | 8 | 69,397 | 69,412 |
| Di- | TA | 10 | 100,045 | 100,064 |
| Di- | TA | 11 | 72,289 | 72,310 |
| Tri- | TTA | 4 | 70,696 | 70,707 |
| Tetra- | AATA | 3 | 54,140 | 54,151 |
| Tetra- | ATAA | 3 | 25,162 | 25,173 |
| Tetra- | ATAG | 3 | 10,865 | 10,876 |
| Tetra- | ATTT | 3 | 69,685 | 69,696 |
| Tetra- | CATA | 3 | 25,129 | 25,140 |
| Tetra- | TACC | 3 | 76,426 | 76,437 |
| Tetra- | TAGA | 3 | 85,926 | 85,937 |
| Penta- | AACAA | 3 | 54,704 | 54,718 |
| Penta- | AAGAA | 3 | 75,527 | 75,541 |

related flowering plants (*Allen et al., 2007*; *Alverson et al., 2010*; *Cho et al., 2004*; *Sloan et al., 2012*), but it is extremely stable in bryophytes (*Liu, Medina & Goffinet, 2014*). The MG of *M. elongata* is 383 bp smaller than the genome of *B. aphylla* (*Liu, Medina & Goffinet, 2014*), which to date is the smallest MG among bryophytes. However, the MG of *M. elongata* contains the same set of genes and a similar genome structure to that of other mosses. The only difference is a pseudogenization of the *nad7* gene.

This locus encodes subunit 7 of NADH dehydrogenase (NDH-1 or complex I of the mitochondrial electron transfer chain) is located on the inner mitochondrial membrane and plays an important role in oxidative phosphorylation process (*Bonen et al., 1994*).

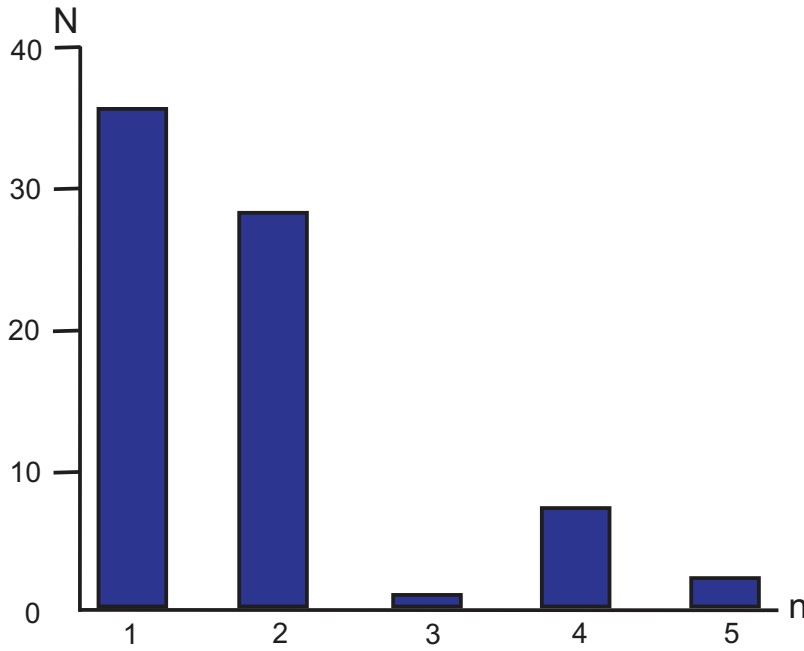

**Figure 3 Simple sequence repeat unit length distribution in *M. elongata* mitogenome.** *n*—the numbers of base pairs ($n = 1, 2, 3, 4$, and 5) in different microsatellite classes. *N*—the number of loci in each SSR category.

NDH-1 is a quite complicated protein complex, consisting of approximately 30–40 subunits (*Kerscher et al., 2008*). The majority of the subunits are encoded in nuclear genome, but several proteins of the complex are specified by mitochondrial genes (*Bonen et al., 1994*).

Although the MGs of the Bryophyta are highly stable in terms of their gene content, there are two other mosses, *B. aphylla* and *T. pellucida* that lack the intact open reading frame (ORF) of the *nad7* gene in their MGs (*Bell et al., 2014*; *Liu, Medina & Goffinet, 2014*). In our study, we found that the exon structure of *nad7* pseudogene of *M. elongata* differs substantially when compared with that of the MGs of *B. aphylla* and *T. pellucida*. Taking into account the close location of the later on the constructed a phylogenetic tree (Fig. 4) and the extremely distant position of *M. elongata* relative to them, the loss of the functionality of the *nad7* gene can be concluded to have occurred at least twice during the evolutionary history of the mosses.

Intact *nad7* genes were found in the MGs of different angiosperms clades (*Adams & Palmer, 2003*) and in representatives of hornworts, lycophytes, ferns and gymnosperms (*Guo et al., 2017*; *Li et al., 2009*; *Xue et al., 2010*). However several exceptions were noted in different evolutionary lineages. Therefore, the absence of a functional *nad7* gene was noted in the MG of *Nicotiana sylvestris* cytoplasmic male sterile (CMS) mutants (*Pla et al., 1995*) and in the lycophyte *Huperzia squarrosa* (*Liu et al., 2012*). In the liverwort *Marchantia polymorpha*, a functional *nad7* gene was transferred from the MG to nucleus, but the pseudogene was preserved in the MG (*Kobayashi et al., 1997*).

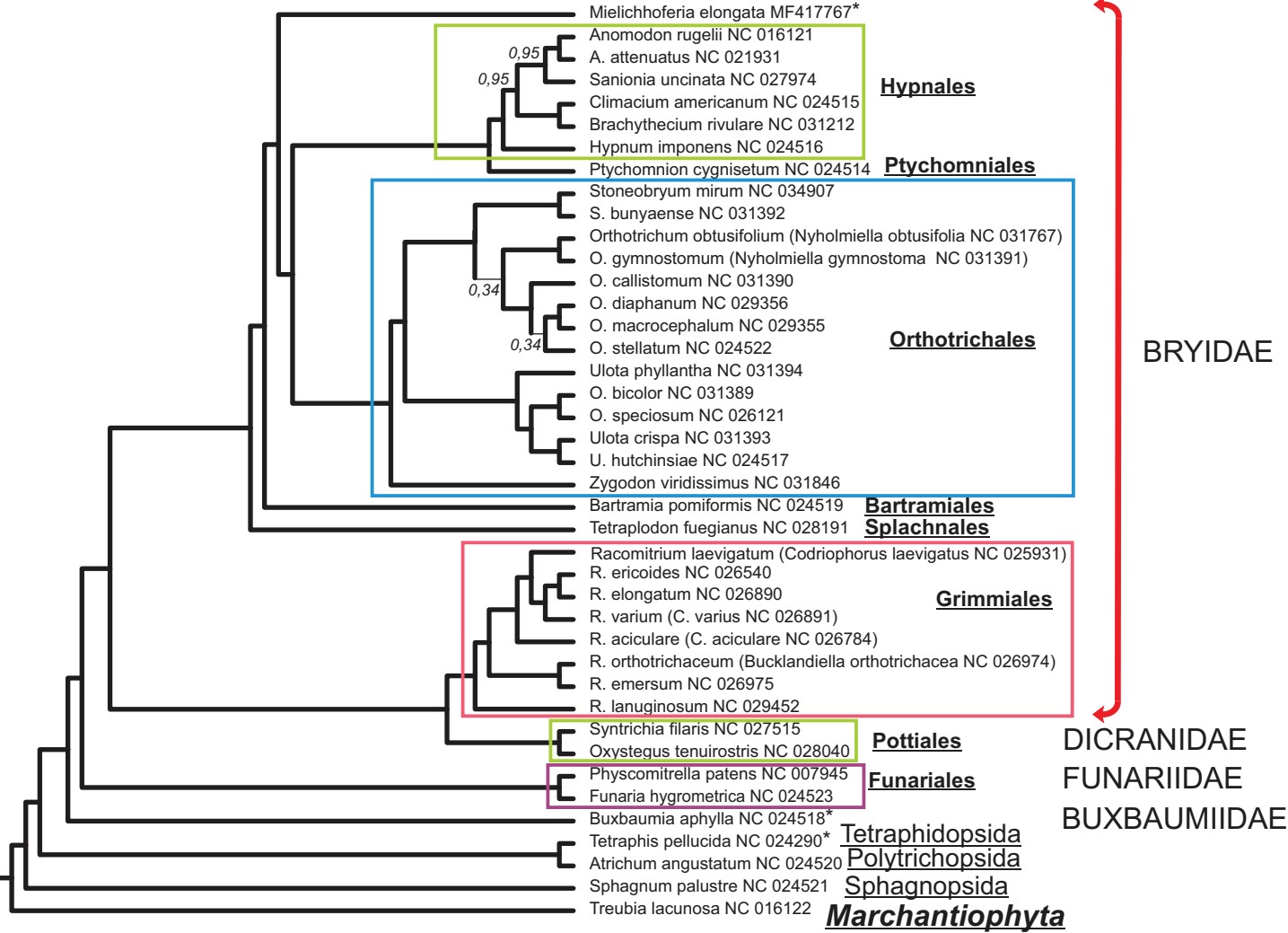

**Figure 4 Bayesian phylogenetic tree of 40 Bryophyta species constructed for 33 mitochondrial protein coding sequences.** The hepatic *Treubia lacunosa* was used as an outgroup. All nodes, except where indicated on the tree, have maximal posterior probability values equal to 1.0. Asterisks indicate taxa with pseudogenization of *nad7*.

Pseudogenization of *nad7* was observed in 11 other liverwort groups, whereas the intact gene was found in *Haplomitrium mnioides* MG (*Groth-Malonek et al., 2007*). This discovery suggested a basal placement of the taxon among liverworts. Overall, pseudogenization of *nad7* may have occurred independently in different unrelated lineages of embryophytes.

A total of 73 SSRs (microsatellites) loci were identified in the MG of *M. elongata*. SSRs are common in plant and animal genomes and could play an important role in gene functioning (*Li et al., 2004*). Besides the occurrence of the SSR loci in nuclear genomes, microsatellite repeats are present in plastids and MGs as well (*Kumar, Kapil & Shanker, 2014*; *Sablok et al., 2015*). However, to date, it is much less known about distribution and functions of microsatellites in bryophyte genomes. SSR loci are usually characterized by

high mutation rate, and therefore actively used as molecular markers in population genetics surveys (*Zalapa et al., 2012*). Molecular markers based on organellar microsatellites have been used successfully for phylogeny reconstruction at the genus taxonomic level and for intraspecific variation analysis (*Ishii, Mori & Ogihara, 2001*; *Nishikawa, Vaughan & Kadowaki, 2005*). The SSR loci revealed in the MG of *M. elongata* could therefore be further investigated to obtain informative markers for using in monitoring programs for *Mielichhoferia* species. That is especially important due to the disruptive character of the habitat area, the rarity of the species, and ongoing habitat damage.

*Mielichhoferia elongata* represents a separate branch on a phylogenetic tree within the Bryidae and is closest to the Hypnales/Ptychomniales/Orthotrichales group. However, the absence of a MG sequence for the Mniaceae and Bryaceae representatives preclude clarification of the taxonomic position of *Mielichhoferia*. The phylogenetic tree depicted in Fig. 4 inferred from 33 mitochondrial CDSes of 40 mosses species with liverwort as an outgroup, is consistent with other reconstructions based on 14–17 plastid genes from 43 moss species representing the major lineages summarized by *Chang & Graham (2013)*, and based on 41 concatenated mitochondrial protein-coding genes from 19 Bryophyta species (*Liu et al., 2014*). Although plant mitochondrial sequences evolve slowly (*Palmer & Herbon, 1988*), phylogenomic analyses can be effective for bryophytes taxa of both lower and higher ranks. Of course, the remarks of *Liu et al. (2014)* and other earlier authors should be kept in minds; namely, that even high support does not guarantee that an inferred phylogeny is approaching the true evolutionary history.

## CONCLUSION

This study provides the complete MG sequence of the "copper moss" *Mielichhoferia elongata* consisting of 100,342 base pairs. It is the smallest known mitochondrial genome among bryophytes and non-parasitic tracheophytes. *M. elongata* is a moss with very specific requirements regarding environmental conditions; in particular, it is mostly confined to heavy metals enriched substrates. Although the MG has the same gene set as that found within previously studied mosses and does not demonstrate any special features associated with high heavy metal tolerance, it lacks a functional *nad7* gene. Based on the phylogeny reconstruction data and exon structure analysis of the gene, it has been deduced, that *nad7* pseudogenization took place independently not once in moss evolution. The phylogenetic tree presented in this study, inferred from the 33 mitochondrial CDS of 41 bryophyte species is consistent with the reconstructions made in earlier studies.

## ACKNOWLEDGEMENTS

We thank reviewers and the editors for their insightful suggestions and comments on the paper. We are grateful to Paul Stothard for assistance in genome map preparation.

### Funding

This work was supported by the Russian Foundation for Basic Research (Nos. 15-04-06027, 18-04-00574, and 15-04-06116). The funders had no role in study design, data collection and analysis, decision to publish, or preparation of the manuscript.

### Grant Disclosures

The following grant information was disclosed by the authors:
Russian Foundation for Basic Research: 15-04-06027, 18-04-00574, and 15-04-06116.

### Competing Interests

The authors declare that they have no competing interests.

### Author Contributions

- Denis V. Goryunov analyzed the data, wrote the paper, prepared figures and/or tables.
- Svetlana V. Goryunova analyzed the data, wrote the paper, reviewed drafts of the paper.
- Oxana I. Kuznetsova performed the experiments.
- Maria D. Logacheva conceived and designed the experiments, performed the experiments, contributed reagents/materials/analysis tools.
- Irina A. Milyutina performed the experiments, analyzed the data.
- Alina V. Fedorova performed the experiments.
- Michael S. Ignatov reviewed drafts of the paper, plants collection and identification.
- Aleksey V. Troitsky conceived and designed the experiments, analyzed the data, contributed reagents/materials/analysis tools, prepared figures and/or tables, reviewed drafts of the paper.

### DNA Deposition

The following information was supplied regarding the deposition of DNA sequences:
The *Mielichhoferia elongata* mitochondrial genome sequence described here is accessible via GenBank accession number MF417767.

### Data Availability

The raw data has been uploaded as Supplemental Dataset Files.

### Supplemental Information

Supplemental information for this article can be found online at http://dx.doi.org/10.7717/peerj.4350#supplemental-information.

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
