# Peer review of "Complete mitochondrial genome sequence of the “copper moss” Mielichhoferia elongata reveals independent nad7 gene functionality loss"

_PeerJ, doi:10.7717/peerj.4350_

## Round 0.1 · original submission · Major Revisions

Two expert reviewers have seen the manuscript and have provided insightful comments. Both reviewers have only minimal concerns about the work performed and about the interpretations of the data. Instead both reviewers feel that the writing is in dire need of attention. I also read the manuscript and agree. The main ideas, motivations, and context are not clearly communicated. This makes it impossible for the reader to understand the rationale behind the research and the implications for the broader field. Additionally, the authors have broken the writing up into many, many short paragraphs, making it nearly impossible to follow their meaning for more than a few sentences. Scientific writing is about clarity of communication and these problems make the current version unsuitable for publication.

I believe that most of these issues can be resolved by having the manuscript reviewed by a native English speaker. I know that I would have a horrible time trying to write in a language not my own and would require such help.

In addition to writing more clearly and providing better context for the results, both reviewers also note a few flaws in communicating the methodology. For example, one reviewer asks for details on the reasons why only some loci were used for the phylogenetic analysis and the other asks for details on the sample preparation. These problems and any others should be addressed when preparing the revision.

Reviewer 1 ·

Basic reporting

The study “Complete mitochondrial genome sequence of the “copper moss” Mielichhoferia elongata reveals independent nad7 gene functionality loss” presents the complete sequence of the mitochondrial genome of the copper moss Mielichhoferia elongata. The results show that the structure of M. elongata mitochondrial genome is pretty similar to that of other moss species, which is in agreement with the fact that the mitochondrial genome of mosses and liverworts evolves at much lower rates than that of flowering plants. Also, the authors characterized a set of SSR loci that could be further used in population genetic and phylogenetic studies. In general, the study presents very valuable results that will help with the phylogenetic reconstruction of the Bryophyte clade (sometimes complicated as demonstrated by the location of the genus Mielichhoferia in different families according to the available literature) among other things. However, the manuscript needs to be revised in depth prior to publication as some major and minor concerns need to be addressed. My major concerns are related to the presentation of the scope of the work and the lack of some relevant information in the materials and methods section.

Experimental design

The scope of the study is not clearly presented. I assumed that lines 58-63 are trying to express the main aim of the work as you can read “So, mitochondrial genome (MG) of Mielichhoferia will be useful both for an understanding of some aspects of heavy metals tolerance mechanism and to find out appropriate taxonomical treatment of the taxa. Besides, mitogenome of M. elongata will facilitate taxonomy and population studies within Mielichhoferia that is important due to the disruptive character of the area, rarity, and habitats damage”. However, first of all, I think that the scope of the study should be expressed more clearly with some statement such as, “the aim of this study is to…”.
Secondly, one of the goals would be “… understanding of some aspects of heavy metals tolerance mechanism…”. I wonder whether there is any evidence (in any species) that points towards the existence of heavy metal tolerance-related genes (or regulatory elements) in plant mitochondrial genomes, if that’s what authors meant with their statement. In case of existing, they should use that information to justify their expectations of understanding heavy metal tolerance mechanisms from mitochondrial genome sequencing. Besides, the only reference made to this issue throughout the rest of the manuscript is in lines 181-182 where the authors point out that “Sequencing did not reveal any special features that can be attributed to high heavy metal tolerance of this moss”. The lack of discussion about this topic makes me think whether “understanding of some aspects of heavy metals tolerance mechanism” should be regarded as one of the aims of this study.
Finally, another goal was “to find out appropriate taxonomical treatment of the taxa”, however, in the discussion section (Lines 210-212) the authors point out that “The absence of MG sequence of Mniaceae and Bryaceae representatives make it impossible to clarify taxonomic position of Mielichhoferia”. If it is impossible to clarify the taxonomic position of Mielichhoferia due to the lack of MG sequences of relevant families, then finding out the appropriate taxonomical treatment of this species should not be one of the aims of the study.
For all these reasons I suggest the authors to revise and rewrite this last part of the introduction in a way that is more related to their results.

“Sample collection and DNA isolation” section: the authors do not provide any explanation about sample storage and preparation before DNA extraction. I think that this is crucial information that cannot be obviated in order to ensure the reproducibility of the experiment and also to assess the comparability of their results with other studies. Which part of the moss did the authors collect for DNA extraction (just tips, whole shoot, etc.)? Was the material cleaned somehow before storage? How was the material stored until DNA extraction (frozen in liquid N/dry ice, dried, etc.)? How much tissue did authors use for DNA extraction? Also, did authors use the DNA extraction kit following the manufacturers’ protocol or did they perform any modification on it? Did they assess the concentration and quality of the DNA? If so, how? The section needs to be completed with all this information.

Validity of the findings

As I stated in the previous section I think that the scope is not well defined and therefore the importance of the findings presented in this work is undermined.

Giving a deep thought to the introduction section and justifying better the importance of the results presented will increase the interest of the reader.

Finally, nad7 pseudogenization is an important feature of the mitochondrial genome of M. elongata, but, as a reader, I’d appreciate some information about its function and how its loss would affect these species that lack a functional nad7.

Additional comments

In general, each paragraph consists of one or two sentences making a total of 2-5 lines per paragraph. This makes the manuscript harder to read. Authors should make an effort gathering small paragraphs referring to the same topic/issue into the same paragraph; e.g. lines 36-49, 50-57, 196-208 could constitute just one paragraph each.

Minor concerns:
1) Abstract: acronyms such as SSR and CDS need to be fully written to ensure the clarity of the summary.
2) Keywords: where it says “Mielichhoferia elongate” it should say “Mielichhoferia elongata”.
3) Line 36: the sentence “Several mosses and hepatics species restricted to substrates enriched with heavy metals” lacks the verb between species and restricted.
4) Line 40: the first time (apart from the abstract section) you use the name of the species this has to be complete: Mielichhoferia elongata. Also, the complete name of M. elongata is not “M. elongata Homsch”, but “Mielichhoferia elongata (Hope & Hornsch) Nees & Hornsch.” According to Tropicos database (www.tropicos.org).
5) Line 50: the authors state that “There are 23 accepted species names of Mielichhoferia Hoppe & Hornsch in the Plant List” but, what does this mean? That there are 23 species under this genus? If that is the point, that is not true because both “The Plant List” and “Tropicos” provide a greater number of species within this genus. If not, please clarify what do you mean in the previous statement.
6) Line 75: what is MHA? Acronyms should be fully written the first time that they are used in the text.
7) Line 108: is there any specific reason why authors chose the MG of Bartramia pomiformis as a reference to annotate M. elongata MG? If so, this should be specified in the text.
8) Line 118: use a dot instead of a comma after M. elongata to shorten the extremely long sentence.
9) Line 136: the authors state that “MG of M. elongata contains 66 genes including genes for 3 rRNAs (rrn18, rrn26, and rrn5), 24 tRNAs, and 40 conserved mitochondrial proteins”, however this would make a total of 67 genes. Taking a look at Table 1 I count 39 protein coding genes, 27 RNA genes which would make a total of 66. Also, I don’t understand why the authors point out that there are 112 genes and one pseudogene in the caption of Table 1, is this a mistake?
10) Figure 2: the figure should be auto explicative so authors should mention in the caption (or legend) that the black filled sections in this figure indicate non-existing exons (or parts of them) in these species.
11) Lines 164-166: is there any reference that supports the criteria for SSR loci identification? If so, this should be added. If not, was this selection based on the criteria regularly used in the literature?
12) Figure 3: this figure needs to be improved. First of all, there is no y-axis label (frequency of SSR classes?) and the line of the x-axis is absent (at least in the PDF version).
13) Line 174: it should be 24,827 instead of 24827.
14) Caption Figure 4: CDS needs to be fully written in order to make the figure auto explicative.
15) Line 184: the sentence “MG size significantly varies even among closely related flowering plants (Allen et al., 2007; Alverson et al. 2010; Cho et al., 184 2004; Sloan et al., 2010; 2012), but extremely stable in bryophytes (Liu, Medina & Goffinet 2014)” needs a verb after but (…but it is extremely…).
16) Line 190: I assume ORF means open reading frame, but this should be specified on the text.
17) Line 213: the authors say that “Phylogenetic tree from 31 mitochondrial CDSes of 40 mosses species” but the tree was built with CDS from 39 moss species and 1 liverwort. This needs to be corrected in the manuscript.
18) Line 228: remove contraction “doesn’t”.
19) References: The reference Sloan et al., 2008 is not cited in the manuscript (in the manuscript I could only find Sloan et al., 2010, 2012).

Reviewer 2 ·

Basic reporting

1. The writing and organization of the introduction section need to be improved, the present version was composed of many sentences rather than well-organized paragraphs. The authors may provide information regarding to the life style/cycle of mosses, the general features and the current research progress in the mitochondrial genome of mosses, and how they differ from other land plant lineages (liverworts, hornworts and vascular plants).
2. There are many grammatical errors throughout the manuscript, e.g. line 23, lines 90-91, line 93, line 97, line 119. The authors may use some language editing services to polish this manuscript.
3. Line 96, line 98, line 100, line 102 and line 105 the parameters of these programs used in this study need to be provided.
4. Line 116, since there are 40 conserved mitochondrial protein-coding genes in mosses, why only 31 of them were used for the phylogenomic analysis? The authors need to justify the selection criteria.
5. Line 22, lines 137-141 and table 1 heading, the numbers were not adding up correctly.

Experimental design

No comment

Validity of the findings

No comment

Additional comments

Goruynov et al. sequenced and assembled the mitochondrial genome of the moss Mielichhoferia elongata. This mitochondrial genome was highly conserved in gene content, genome size and structure except a functional loss of the nad7 gene. Detailed exon structure comparison and phylogenetic analysis show the pseudolization of nad7 in M. elongata was different and independent of that in the two other mosses Buxbaumia aphylla and Tetraphis pellucida. Overall this study is mainly descriptive.

---

## Round 0.2 · Minor Revisions

Please address the remaining minor concerns of the reviewer and return at your earliest convenience.

Reviewer 2 ·

Basic reporting

no comment

Experimental design

no comment

Validity of the findings

no comment

Additional comments

The writing of the manuscript has been improved, I only have some minor comments below:

1. Line 87, typo? “2 mkg DNA”.
2. Line 163, spelling error, “M. elongate” should be “M. elongata”.
3. Line 120-122, only 23 out of the 31 protein-coding genes were listed. And since nad7 is a pseudo-gene in some mosses, why it was used for the phylogenetic analysis?
4. Fig. 1 looks messy, I suggest the authors take a look at some mitochondrial genome maps in other publications.
5.Table 1 heading, typo, “60 genes”.

---

## Round 0.3 · accepted · Accept

Thank you for your attention to detail.